# Early Growth Assessment of *Lolium perenne* L. as a Cover Crop for Management of Copper Accumulation in Galician Vineyard Soils

Raquel Vázquez-Blanco [1,2], Manuel Arias-Estévez [1,2], David Fernández-Calviño [1,2]
and Daniel Arenas-Lago [1,2,*]

1   Departamento de Bioloxía Vexetal e Ciencias do Solo, Area de Edafoloxía e Química Agrícola,
    Facultade de Ciencias, Universidade de Vigo, As Lagoas s/n, 32004 Ourense, Spain;
    raquel.vazquez.blanco@uvigo.gal (R.V.-B.); mastevez@uvigo.es (M.A.-E.); davidfc@uvigo.gal (D.F.-C.)
2   Instituto de Agroecoloxía e Alimentación (IAA), Campus Auga, Universidade de Vigo, 32004 Ourense, Spain
*   Correspondence: darenas@uvigo.es

**Abstract:** This study investigates the potential use of *Lolium perenne* L. as a cover crop to improve vineyard soils with varying levels of copper (Cu). Cu-based fungicides are commonly used to control fungal diseases in vineyards, but their accumulation in soils poses environmental risks. This study aims to address this issue by evaluating the influence of soil properties on Cu availability and *L. perenne* growth. A total of 42 vineyard soils from different Designations of Origin (D.O.s) in Galicia were sampled and their physicochemical properties were analyzed. The results showed most soils exceeded recommended Cu limits due to fungicide applications. Pot experiments were conducted to assess *L. perenne* growth and Cu accumulation. *L. perenne* biomass did not vary significantly with total soil Cu content, indicating that other factors such as organic matter and cation exchange capacity were more important for plant growth. While *L. perenne* showed Cu tolerance, its aerial Cu accumulation was inversely correlated with available Cu. This study provides insight into the potential of *L. perenne* as a cover crop for sustainable vineyard management and soil improvement and emphasizes the importance of considering Cu accumulation from fungicide applications.

**Keywords:** copper; fungicide; phytosanitary; cover crop; vineyard; organic viticulture; sustainable agriculture

## 1. Introduction

In recent decades, the viticulture sector in Galicia (Spain) has gained considerable importance. According to global data from the Designations of Origin (D.O.s) in Galicia, the hectares dedicated to vine monoculture, as well as grape and wine production, have increased significantly in recent years, reaching an area of nearly 10,000 hectares dedicated to viticulture, with an annual grape yield of nearly 50,000 tons [1]. Viticulture in Galicia is organized into five Designations of Origin (D.O.s), which encompass regions where vineyards hold greater importance. Each D.O. in Galicia is characterized by the cultivation of different grape varieties, adapted to the specific edaphoclimatic conditions of each region [2–6]

Galicia's vineyard soils display significant edaphological variation depending on the viticultural area and geological features of each D.O. Nevertheless, they are mostly recognized for being acidic, shallow, and nutrient-deficient as a result of leaching caused by precipitation. The climate in Galicia's viticultural areas is generally categorized as oceanic, producing mild winters and cool summers. Rainfall is distributed throughout the year [7]. More specifically, the Rías Baixas D.O. experiences an Atlantic climate with mild temperatures and high humidity. Similarly, the Ribeiro D.O. also has an Atlantic climate with Mediterranean influence and experiences higher temperatures during hot

and dry summers. The precipitation concentrated in winter and spring stimulates the vegetative cycle of the vines. The Ribeira Sacra D.O. features an Atlantic climate with some continental influence. The region experiences extreme temperatures including cold winters and hot summers. However, rainfall is infrequent, leading to potential water stress for the vines. The Valdeorras D.O. has a continental climate with Atlantic influence, marked by low precipitation, cold winters, and hot summers. The region has a marked thermal oscillation between day and night. The Monterrei D.O. has a Mediterranean climate with Atlantic influence, presenting variations depending on the altitude and orientation of the slopes. Overall, the Monterrei D.O. has mild winters and hot, dry summers. However, the region experiences scarce precipitation. The edaphoclimatic characteristics of these regions favor the occurrence of fungal diseases [8]. As a result, there is intensive application of phytosanitary treatments, primarily copper-based fungicides, to maintain the productivity of the sector [9,10].

Copper (Cu) is an essential element for plant growth as it is a necessary micronutrient in plant metabolism. Even though the concentration of Cu in the dry tissue of plants is typically low (<0.01%), it is vital for essential functions, including photosynthesis, protein synthesis, and enzyme activation [11]. Nevertheless, elevated concentrations of Cu in the soil can prove harmful to plants. Soil Cu levels exceeding 100 mg kg$^{-1}$ could affect plant growth and development and pose a threat to soil quality [12]. Copper in the soil mainly exists in the form of divalent cations in the soil solution or interacts with organic matter to form organometallic complexes, along with soil mineral components, such as aluminum, iron, and manganese oxihydroxides, and clays [13]. Copper mobility in the soil depends on the pH; it tends to form insoluble and less available compounds in alkaline soils while being more available to plants in acidic or neutral soils, potentially leading to toxicity issues [14,15].

Copper toxicity in plants causes various symptoms, including leaf darkening, chlorosis, and root malformation. Moreover, different harmful effects have been observed as a result of copper toxicity in plants. These effects include damage to plant tissues, abnormal elongation of root cells, alterations in membrane permeability, lipid peroxidation in chloroplasts, inhibition of electron transport in photosynthesis, immobilization of Cu in cell walls, and damage to genetic material. Despite this, certain plant species have demonstrated a greater capacity for resistance and adaptation to the presence of copper, while others are more sensitive and can experience negative impacts even at lower concentrations [15]. As a result, it is essential to assess the availability and toxicity of Cu in the soil to ensure proper crop management, as well as prevent negative impacts on plant health and soil quality [16].

The presence of Cu in soil may have natural origins, but contamination and toxicity of Cu in soils and plants are primarily due to anthropogenic activities such as mining, the timber industry, metallurgy, and excessive use of fertilizers and pesticides in the agricultural sector [17]. Although Cu is an essential nutrient for plant growth, it is also used in agriculture as a component of fungicides and fertilizers to control plant diseases and enhance productivity, particularly in crops such as vineyards [18].

Cu-based phytosanitary products have been widely used since the mid-nineteenth century to protect vineyards due to their high susceptibility to fungal diseases, such as mildew (*Plasmopara vitícola*) and powdery mildew (*Uncinula necátor*) [19]. Chemical substances with antifungal properties, containing Cu, have been used to combat these diseases. The most commonly used compounds include Bordeaux mixture (Ca(OH)$_2$ + CuSO$_4$), copper oxychloride (Cu$_2$(OH)$_3$Cl), and commercial brand compounds. These substances provide an effective defense against vine diseases while maintaining plant health and vineyard productivity. These compounds are frequently applied, leading to Cu accumulation that often exceeds the 100 mg kg$^{-1}$ threshold in vineyard soil [12,13]. This jeopardizes soil and environmental health by creating a potential risk of pollution.

The application of fungicides containing Cu in vineyard protection can significantly affect the soil in which they are applied. Studies have shown that Cu present in such fungicides can enter the soil by directly depositing from treated leaves or by washing due

to precipitation [20]. When Cu enters the soil, it tends to accumulate in the top layer of the soil due to its high affinity for organic matter [21]. The accumulation of Cu in the soil can have harmful effects on future crops established in the same area or on crops established in nearby areas. The high copper content in the soil can be toxic to plants, including wild plants or cover crops grown in vineyards [22]. This can potentially affect the plants due to the presence of high levels of Cu in the soil [23]. The concurrent presence of copper and cationic herbicides in the soil can intensify the negative effects on soil quality and ecosystem health. This interaction can cause copper retention and increase the risk of environmental contamination, particularly through the leaching of these compounds [24]. Several studies have reported that excessive or improper use of Cu products in agriculture can increase Cu concentrations in the soil and potentially harm plants and the environment [25–27]. Regarding soils, where Cu-based fungicides are commonly used, the long-term cumulative effects of this metal on the soil are related to: (i) problems with the uptake of essential nutrients due to changes in the ionic balance in roots [28]; (ii) reduction in biodiversity and microbial activity, which can affect soil biological processes, including organic matter decomposition, nitrogen fixation, and nutrient availability to plants [21]; (iii) alteration of soil structure, causing compaction and reduced porosity, which can affect water infiltration, soil aeration, and nutrient retention capacity [29]; (iv) contamination of groundwater as a result of Cu leaching into deeper soil layers and eventually reaching groundwater [30]; (v) and accumulation in fruits and wine products, which can lead to the presence of Cu residues in wine products, raising concerns for human health and wine quality standards [26]. Therefore, proper management during fungicide application and the adoption of sustainable agricultural practices are required to minimize the impact of Cu on soils and ensure the long-term health of vineyards and the quality of wine products.

Over the years, vineyard management techniques have undergone significant changes. Initially, economic and production criteria were the main drivers of changes in vineyard management systems, before a holistic assessment of the environmental and agricultural impacts associated with vineyard operations began [31]. In the context of the Green Revolution that began in the mid-20th century, there was an increase in global agricultural productivity due to the introduction of new techniques, technologies, and inputs into agro-ecosystems. This led to the establishment of intensive agriculture as we know it today. In vineyards, the use of phytosanitary products contributed significantly to the stabilization of productive yields and grape quality over time. However, this intensification has not been without negative environmental and human health impacts, including soil degradation and loss of fertility, groundwater contamination, pest proliferation, excessive use of agricultural inputs, and loss of biodiversity [32,33]. In response to these challenges, alternatives based on the principles of agroecology have emerged in recent years, which are based on ecological principles and biological interactions in agrosystems, with the aim of making agricultural processes sustainable. This includes pest control, appropriate fertilization, and environmental protection in all dimensions [34,35].

In recent years, the implementation and advancement of organic viticulture have been fundamental steps to improve soil health and vineyard quality [36,37], mainly through sustainable management practices such as (i) minimum tillage to preserve soil structure, reduce erosion, and promote microbiological activity; (ii) nutrient application and management to determine the nutritional needs of the vine and balanced fertilizer use; (iii) moisture management to optimize water use and prevent soil compaction; (iv) integrated pest and disease control: implementation of control strategies and regular monitoring of pest populations and use of environmentally friendly biological control methods; and (v) use of cover crops to improve soil quality, maintain biodiversity, and reduce erosion.

Specifically, the use of cover crops in vineyards is an emerging agricultural strategy aimed at improving the sustainability of the viticultural system. Cover crops are the planting of selected plant species between rows of vines to protect the soil, increase its fertility, and regulate weed competition [38]. Cover crops can be categorized into natural cover and planted cover. Natural covers consist of native plant species that grow spontaneously

in the vineyard environment, are generally well adapted to local conditions, contribute to the soil characteristics and biodiversity of the vineyard ecosystem, and provide valuable ecosystem services [39]. On the other hand, planted cover involves the selection and planting of specific plant species for agronomic purposes. Planted covers may include legumes such as clover or vetch, grasses such as rye or barley, or mixtures of species that provide complementary benefits [40]. Both types of cover crops offer benefits, and their selection will depend on the specific viticultural management objectives and environmental conditions of each vineyard. Regardless of their type, cover crops offer several characteristics that make them beneficial to the viticultural system. Cover crops protect the soil from climate-induced erosion, as both the roots and above-ground parts act as a physical barrier to prevent soil and nutrient loss; they help improve soil structure by promoting the formation of aggregates that increase porosity and water-holding capacity; they add plant material to the soil, increasing organic matter and soil biodiversity; they compete with weeds, reducing their growth and minimizing the need for herbicide application; they can promote soil nitrogen fixation; they contribute to biological pest and disease control, reducing the need for pesticides; and they influence the vineyard microclimate, reducing temperature fluctuations and maintaining soil moisture, which can promote optimal vine development and grape quality [41]. However, the use of cover crops also presents challenges and inconveniences that must be properly addressed to maximize their effectiveness and benefits. Cover crops can compete with vines for resources such as water and nutrients, especially in the early stages of growth. Therefore, the selection of cover crop species is a critical factor to ensure they do not become invasive weeds in the vineyard. In addition, managing the height and density of cover crops is essential to balance their interaction with vines and maximize their agronomic performance [42].

Among the species with potential for use as cover crops in vineyards is *Lolium perenne* L. This species, commonly known as perennial ryegrass or English ryegrass, is an herbaceous plant belonging to the grass family (Poaceae). It has a fibrous root system and erect stems that can reach heights of 30 to 100 cm. It has high germination and vigor, which facilitate its expansion. *Lolium perenne* is widely used due to its various applications, including its use as a high-quality forage species used in livestock feed; for lawns due to its resistance and rapid germination; in seed mixtures used for re-vegetation of eroded or degraded areas; for erosion control on slopes and areas with excessive steepness; as a phytoremediation species in soils contaminated with potentially toxic elements, thanks to its ability to accumulate metals in its tissues; and as a cover crop to improve soil conditions and control erosion [43–48].

In the present study, the use of *L. perenne* as a cover crop in vineyard soils from different Denominations of Origin (D.O.s) in Galicia was evaluated from an agroecological perspective. This species can provide soil protection against environmental factors that cause erosion, improve soil structure, and sequester part of the Cu present in the soil as a result of the continuous use of fungicides. Therefore, the main objectives of this research are: (i) to study the influence of physicochemical soil characteristics in the availability of Cu accumulated in vineyard soils as a consequence of fungicide application; (ii) to know the possible influence of available Cu in the vineyard soils on the early growth of *L. perenne*, and (iii) to assess the capacity of *L. perenne* to be used as a cover crop and to improve the conditions of Galician vineyard soils under stress caused by high Cu contents.

## 2. Materials and Methods

### 2.1. Study Area and Selection and Sampling of Vineyard Soils

In this study, vineyard soils were selected from the five Designations of Origin (D.O.s) in Galicia: Rías Baixas, Ribeiro, Ribeira Sacra, Monterrei, and Valdeorras. The vineyards were selected based on different criteria: ecological, socio-economic, topographical, and climatic, to ensure a wide variability within the study area. All the vineyards studied suffer from high humidity, which requires intensive application of Cu-based fungicides, mainly in the form of a Bordeaux mixture and copper oxychloride, among others.

A total of 42 vineyard soils were sampled from 34 different vineyards in the five D.O.s mentioned above (Rías Baixas: 5 samples; Ribeiro: 12 samples; Ribeira Sacra: 10 samples; Valdeorras: 11 samples; and Monterrei: 4 samples) (Table 1). The numbering of the soils was based on the total Cu content in each soil, with soil 1 having the lowest Cu content and soil 42 having the highest. In addition, the soils were distinguished between those sampled under the vine strain (VS) (20 samples) and those sampled in the alleys between the vine rows (VR) (22 samples).

**Table 1.** Location and origin of the sampled vineyard soils.

| Soil | Sampling Area | D.O. | Municipality | Soil | Sampling Area | D.O. | Municipality |
|------|--------------|------|--------------|------|--------------|------|--------------|
| 1 M | VR | Monterrei | Vilardevós | 22 RS | VR | Ribeira Sacra | Sober |
| 2 RB | VS | Rías Baixas | Salvaterra de Miño | 23 M | VR | Monterrei | Verín |
| 3 RS | VR | Ribeira Sacra | A Pobra de Trives | 24 RS | VR | Ribeira Sacra | Parada de Sil |
| 4 M | VR | Monterrei | Monterrei | 25 RS | VR | Ribeira Sacra | A Teixeira |
| 5 M | VS | Monterrei | Oímbra | 26 V | VS | Valdeorras | Vilamartín de Valdeorras |
| 6 V | VR | Valdeorras | O Barco de Valdeorras | 27 RS | VS | Ribeira Sacra | Parada de Sil |
| 7 RS | VS | Ribeira Sacra | A Pobra de Trives | 28 RB | VR | Rías Baixas | Arbo |
| 8 RB | VR | Rías Baixas | Tomiño | 29 R | VR | Ribeiro | Sober |
| 9 RB | VS | Rías Baixas | Meis | 30 RS | VS | Ribeira Sacra | Sober |
| 10 V | VR | Valdeorras | Vilamartín de Valdeorras | 31 RS | VS | Ribeira Sacra | Sober |
| 11 V | VS | Valdeorras | Petín | 32 R | VS | Ribeiro | Beade |
| 12 R | VS | Ribeiro | Ribadavia | 33 RS | VS | Ribeira Sacra | Parada de Sil |
| 13 R | VR | Ribeiro | Ribadavia | 34 R | VR | Ribeiro | Arnoia |
| 14 R | VR | Ribeiro | Cenlle | 35 V | VS | Valdeorras | A Rúa |
| 15 V | VS | Valdeorras | Rubiá | 36 R | VR | Ribeiro | Toén |
| 16 V | VR | Valdeorras | Rubiá | 37 R | VR | Ribeiro | Toén |
| 17 V | VR | Valdeorras | Rubiá | 38 R | VS | Ribeiro | Sober |
| 18 RB | VS | Rías Baixas | Arbo | 39 V | VS | Valdeorras | Rubiá |
| 19 RS | VS | Ribeira Sacra | Sober | 40 V | VS | Valdeorras | Vilamartín de Valdeorras |
| 20 R | VR | Ribeiro | Sober | 41 R | VR | Ribeiro | Toén |
| 21 V | VS | Valdeorras | O Barco de Valdeorras | 42 R | VR | Ribeiro | Toén |

D.O.: Denomination of origin; RB: Rías Baixas; RS: Ribeira Sacra; R: Ribeiro; M: Monterrei; V: Valdeorras; VR: vineyard row; VS: vineyard strain; N: North; W: West.

At each sampling point, three subsamples (0–20 cm depth) were collected using an Edelman auger. The soil subsamples from each sampling point were mixed and homogenized to form a composite soil sample, which was stored in polyethylene bags. At the laboratory, the composite samples were air dried, sieved through a 2 mm mesh, and stored in polyethylene bottles until analysis.

### 2.2. Soil Characterization

Soil pH was determined both in deionized water and in a 0.1 M KCl solution at a soil/water ratio of 1:2.5 (*w/v*) [49]. Particle size analysis and soil texture were determined by the Day method [50]. The total carbon and nitrogen contents in the soil were determined using a Thermo Flash EA 1112 elemental analyzer. The sample was completely combusted in a reactor with a high-temperature oxidizing catalyst (900 °C). The elemental gases produced were analyzed by chromatography and a high-sensitivity thermal conductivity detector to determine carbon and nitrogen. The organic carbon content was determined by incinerating the sample in a muffle furnace at 450 °C for 4 h. The organic carbon content was calculated as the difference between the weight of the sample before and after ashing. The organic matter content was estimated by multiplying the total carbon content by the factor 1.724 proposed by Van Bemmelen for carbon contents below 5.8% and by 2 for higher contents. The dissolved organic carbon content was determined using a Shimadzu TOC-V CSH/CSN analyzer [51]. The effective cation exchange capacity (eCEC) was determined as the sum of $Ca^{2+}$, $Mg^{2+}$, $Na^+$, and $K^+$ (exchangeable bases) and $Al^{3+}$ in the exchange complex [52]. Cation contents were determined by atomic absorption spectroscopy (Thermo Solaar Series M, Thermo Fisher Scientific Inc., Waltham, Massachusetts, USA). To determine the $Al^{3+}$ content in the exchange complex, a preliminary extraction with 1 M KCl was performed and the $Al^{3+}$ content was determined by acid-base titration with 0.1 M NaOH. The results were expressed as $cmol_{(+)}$ $kg^{-1}$ soil.

### 2.3. Total and Available Cu Content in Vineyard Soils

The total Cu content in vineyard soils was determined by acid extraction with aqua regia ($HCl:HNO_3$ ratio 3:1 *v/v*) using Teflon reactors in a microwave oven (190 °C and 9 bars, time 45 min). The Cu content was determined by atomic absorption spectroscopy (Thermo Solaar Series M, Thermo Fisher Scientific Inc.).

For the available Cu content in vineyard soils, 3 g of vineyard soils from each pot was weighed into a polyethylene tube and 30 mL of a solution of acetic, lactic, citric, malic, and formic acid was added for a total acid concentration of 10 mM and a molar ratio of 4:2:1:1:1. Soils with the acid mixture were shaken for 16 h, centrifuged, and filtered [53]. The extracts were analyzed by atomic absorption spectroscopy (Thermo Solaar Series M, Thermo Fisher Scientific Inc.) to determine the available Cu content.

For this purpose, the total concentration of the acids was 10 mM and the molar ratio was 4:2:1:1:1:1

The extraction efficiency ($E_xE_f$) of the acid mixture was determined to evaluate the percentage of available Cu relative to the total Cu content in the vineyard soils. The $E_xE_f$ was calculated using the following equation:

$$E_xE_f = 100 \cdot \frac{Cu_a}{Cu_t} \qquad (1)$$

where $Cu_a$ and $Cu_t$ are available Cu (mg $kg^{-1}$) and total Cu (mg $kg^{-1}$) content, respectively, for each vineyard soil.

### 2.4. Plant Material and Pot Experiments

*Lolium perenne* seeds were used for the pot experiments. Seedlings were grown in seedbeds, and 80 g of each of the vineyard soil samples listed in Table 1 were added to individual pots. For each soil, 5 pots were used. Then, 5 g of *L. perenne* seeds were added to each pot to ensure proper burial in the soil. After sowing, the pots were placed in a climate chamber with a light–dark cycle of 16 h and 8 h, respectively, and a constant temperature of 25 °C. For one month, the pots were irrigated with distilled water every two days to maintain the soil's moisture retention capacity. At the end of the month, the above-ground parts of each plant were harvested and the fresh biomass of *L. perenne* in each pot was

recorded. The samples were then wrapped in aluminum foil and dried in an oven at 40 °C for 72 h. Finally, the dried samples were weighed to determine the dry biomass.

### 2.5. Determination of Cu in L. perenne Shoots

The dried shoot plant samples from each pot were digested with concentrated $HNO_3$ using the following procedure: 0.2 g of the plant sample was collected and placed in Pyrex glass tubes, together with 7.5 mL of concentrated $HNO_3$. The samples were allowed to contact the $HNO_3$ for 12 h at room temperature. The tubes were then sealed and placed in a digestion block at 105 °C for 2 h. After the digestion period, the tubes were allowed to cool for 1 h before repeating the process for another 2 h. The contents of the tubes were then filtered and transferred to polyethylene tubes filled to the mark with distilled water. The Cu content was determined by atomic absorption spectrometry (Thermo Solaar Series M, Thermo Fisher Scientific Inc.).

The bioaccumulation factor (BF) was determined to evaluate the relationship between the Cu content in the aerial part of *L. perenne* and the available Cu content in the vineyard soils. The BF was calculated using the following equation:

$$ \text{BF} = \frac{\text{Cu content in shoots of } L.\ perenne}{\text{available Cu content in vineyard soil}} \cdot 100 \tag{2} $$

### 2.6. Statistical Analysis

All analyses of soils and plants were performed in triplicate. Statistical analysis of the data was performed using Past4.11 software (Oslo, Norway) for Windows. A bivariate Pearson correlation analysis was performed between the soil characteristics and the total and available Cu contents, as well as the parameters determined in the pot experiments with *L. perenne*. To interpret the correlation analysis, the criterion, where absolute values of R between 0.3–0.5 represent a medium effect and R ≥ 0.5 represents a large effect, was used [54]. Principal component analysis (PCA) was performed on the dataset to identify possible relationships between soil characteristics, total and available Cu content in soils, Cu content in the aerial part of *L. perenne*, and its biomass. For the PCA, soil samples were separated by sampling zone (vine strain or vine row) and grouped by designation of origin (D.O.). The correlation matrix was used and group differences (D.O.) were taken into account in the analysis.

## 3. Results and Discussion

### 3.1. General Characterization of the Vineyard Soils

The physico-chemical characteristics of the studied vineyard soils of the different Galician D.O.s are shown in Table 2. A high variability in soil composition was observed. Although most of the soils presented a sandy loam texture, significant variations in silt and clay content were found, resulting in the presence of silt loam, sandy clay loam, and loam textures in general. The presence of clay in the soil is particularly relevant due to its fundamental role in Cu retention [13]. Clays can retain cations, including Cu, within their crystalline structure or in cation exchange sites. Therefore, the prevalence of textures with a low clay content in vineyard soils may imply a lower capacity to bind Cu in the mineral fraction of the soil. This means that Cu could be more available in a free or easily exchangeable form, which could affect its availability to vines and potentially influence their growth [23]. In addition to its effect on Cu retention, soil texture also plays a critical role in its ability to retain water. Textures with higher proportions of clay and silt have a greater capacity to hold water than sandy textures. This is because finer particles, such as silt and clay, have a higher specific surface area and therefore a greater capacity to hold water. Consequently, soil texture has a significant impact on the amount of water available for proper vine growth and development, as it determines the water reserve available to the roots [55].

**Table 2.** General characteristics of vineyard soils (average values).

| Soil | S.a. | pH$_{H2O}$ | pH$_{KCl}$ | C(%) | OM | N | C/N | DOC | DIC | Ca$^{2+}$ | Mg$^{2+}$ | K$^+$ | Na$^+$ | Al$^{3+}$ | eCEC | Sand | Silt | Clay | Texture |
|---|---|---|---|---|---|---|---|---|---|---|---|---|---|---|---|---|---|---|---|
| | | | | | (%) | | - | mg kg$^{-1}$ | | | | cmol$_{(+)}$ kg$^{-1}$ | | | | | (%) | | |
| 1 M | VR | 5.35 | 3.31 | 2.62 | 4.52 | 0.21 | 12.61 | 20.36 | 0.09 | 0.06 | 0.44 | 0.57 | 0.05 | 2.27 | 3.38 | 33.06 | 49.94 | 17.01 | silt loam |
| 2 RB | VS | 7.37 | 5.42 | 4.48 | 7.72 | 0.27 | 16.77 | 26.20 | 0.79 | 13.48 | 1.40 | 0.79 | 0.09 | 0.00 | 15.75 | 46.54 | 35.87 | 17.59 | loam |
| 3 RS | VR | 6.37 | 4.03 | 1.62 | 2.79 | 0.12 | 13.07 | 25.16 | 0.40 | 0.26 | 0.56 | 0.62 | 0.03 | 0.37 | 1.83 | 69.68 | 16.57 | 13.74 | sandy loam |
| 4 M | VR | 6.46 | 4.18 | 1.07 | 1.84 | 0.08 | 12.63 | 16.12 | 0.50 | 0.18 | 0.70 | 0.44 | 0.05 | 0.00 | 1.38 | 72.99 | 14.27 | 12.75 | sandy loam |
| 5 M | VS | 4.86 | 2.94 | 1.73 | 2.98 | 0.13 | 13.31 | 27.85 | 0.19 | 0.14 | 0.73 | 0.75 | 0.06 | 1.69 | 3.37 | 72.09 | 15.06 | 12.85 | sandy loam |
| 6 V | VR | 8.02 | 7.32 | 2.84 | 4.90 | 0.37 | 7.70 | 28.58 | 11.47 | 18.00 | 0.76 | 0.38 | 0.04 | 0.00 | 19.18 | 44.82 | 36.10 | 19.08 | loam |
| 7 RS | VS | 5.76 | 3.75 | 2.54 | 4.38 | 0.20 | 12.68 | 35.29 | 0.22 | 0.13 | 0.44 | 0.59 | 0.04 | 1.34 | 2.54 | 69.68 | 16.57 | 13.74 | sandy loam |
| 8 RB | VR | 7.71 | 5.88 | 3.04 | 5.24 | 0.18 | 17.04 | 18.74 | 3.63 | 11.56 | 1.07 | 0.72 | 0.31 | 0.00 | 13.65 | 55.42 | 28.63 | 15.95 | sandy loam |
| 9 RB | VS | 6.87 | 4.69 | 3.78 | 6.52 | 0.29 | 13.27 | 14.92 | 0.51 | 7.64 | 1.15 | 0.96 | 0.10 | 0.02 | 9.87 | 64.65 | 19.03 | 16.32 | sandy loam |
| 10 V | VR | 5.45 | 2.96 | 1.30 | 2.24 | 0.11 | 11.91 | 17.70 | 0.01 | 0.10 | 0.40 | 1.05 | 0.03 | 0.89 | 2.48 | 32.52 | 52.00 | 15.49 | silt loam |
| 11 V | VS | 7.73 | 6.09 | 3.94 | 6.79 | 0.14 | 28.23 | 23.70 | 2.06 | 6.16 | 1.25 | 0.57 | 0.03 | 0.00 | 8.01 | 65.45 | 21.25 | 13.30 | sandy loam |
| 12 R | VS | 4.87 | 3.75 | 0.67 | 1.16 | 0.05 | 13.07 | 19.34 | 0.02 | 0.32 | 0.84 | 0.53 | 0.42 | 2.23 | 4.34 | 70.15 | 16.20 | 13.65 | sandy loam |
| 13 R | VR | 4.83 | 3.81 | 0.62 | 1.07 | 0.05 | 12.53 | 18.14 | 0.00 | 0.29 | 0.66 | 0.52 | 0.44 | 1.93 | 3.84 | 70.15 | 16.20 | 13.65 | sandy loam |
| 14 R | VR | 5.68 | 4.27 | 1.32 | 2.28 | 0.10 | 13.78 | 18.87 | 0.07 | 0.10 | 0.33 | 0.66 | 0.03 | 0.70 | 1.81 | 61.47 | 24.14 | 14.40 | sandy loam |
| 15 V | VS | 5.77 | 3.99 | 1.39 | 2.40 | 0.15 | 9.50 | 25.07 | 0.05 | 0.42 | 0.42 | 0.61 | 0.03 | 1.12 | 2.60 | 49.56 | 24.31 | 26.13 | sandy clay loam |
| 16 V | VR | 6.48 | 4.67 | 1.45 | 2.50 | 0.16 | 9.06 | 34.04 | 0.52 | 0.50 | 0.44 | 0.47 | 0.03 | 0.00 | 1.44 | 49.56 | 24.31 | 26.13 | sandy clay loam |
| 17 V | VR | 6.48 | 4.26 | 1.63 | 2.81 | 0.16 | 10.19 | 30.21 | 0.24 | 4.68 | 1.42 | 0.45 | 0.06 | 0.00 | 6.61 | 39.11 | 43.88 | 17.01 | loam |
| 18 RB | VS | 7.75 | 5.90 | 2.62 | 4.52 | 0.22 | 11.93 | 35.11 | 2.70 | 11.76 | 1.17 | 0.71 | 0.04 | 0.21 | 13.89 | 52.80 | 31.60 | 15.59 | sandy loam |
| 19 RS | VS | 5.70 | 4.93 | 3.82 | 6.59 | 0.24 | 15.62 | 115.21 | 0.47 | 4.95 | 2.58 | 0.84 | 0.01 | 0.33 | 8.38 | 61.49 | 26.47 | 12.04 | sandy loam |
| 20 R | VR | 5.55 | 5.33 | 1.96 | 3.38 | 0.17 | 13.10 | 54.36 | 1.93 | 4.77 | 1.66 | 0.94 | 0.01 | 0.34 | 7.38 | 62.16 | 23.10 | 14.74 | sandy loam |
| 21 V | VS | 7.78 | 6.47 | 1.74 | 3.00 | 0.14 | 12.57 | 34.53 | 6.91 | 12.64 | 0.76 | 0.55 | 0.04 | 0.64 | 14.62 | 53.30 | 32.06 | 14.64 | sandy loam |
| 22 RS | VR | 4.61 | 4.49 | 3.75 | 6.47 | 0.23 | 15.99 | 101.82 | 0.34 | 4.45 | 1.45 | 0.68 | 0.01 | 0.97 | 7.56 | 56.97 | 31.71 | 11.32 | sandy loam |
| 23 M | VR | 6.77 | 4.81 | 1.50 | 2.59 | 0.12 | 12.84 | 20.46 | 2.81 | 0.47 | 0.52 | 0.67 | 0.04 | 0.63 | 2.32 | 54.81 | 30.78 | 14.40 | sandy loam |
| 24 RS | VR | 6.19 | 4.63 | 2.97 | 5.12 | 0.21 | 14.15 | 35.18 | 1.33 | 7.20 | 0.55 | 0.54 | 0.07 | 0.00 | 8.35 | 68.25 | 17.84 | 13.91 | sandy loam |
| 25 RS | VR | 6.19 | 3.81 | 1.95 | 3.36 | 0.14 | 14.19 | 28.93 | 0.15 | 0.28 | 0.51 | 0.67 | 0.03 | 0.47 | 1.95 | 61.92 | 24.12 | 13.96 | sandy loam |
| 26 V | VS | 7.29 | 5.49 | 2.76 | 4.76 | 0.22 | 12.56 | 40.78 | 4.34 | 11.80 | 1.25 | 1.25 | 0.04 | 0.00 | 14.35 | 30.96 | 50.04 | 19.00 | silt loam |
| 27 RS | VS | 7.60 | 5.98 | 1.87 | 3.22 | 0.14 | 13.54 | 31.28 | 4.08 | 6.80 | 1.87 | 1.80 | 0.13 | 0.05 | 10.65 | 65.83 | 19.52 | 14.66 | sandy loam |
| 28 RB | VR | 6.47 | 4.08 | 0.90 | 1.55 | 0.10 | 9.50 | 20.18 | 0.03 | 0.12 | 0.25 | 0.66 | 0.02 | 0.15 | 1.20 | 35.17 | 47.55 | 17.28 | loam |
| 29 R | VR | 5.29 | 4.70 | 1.35 | 2.33 | 0.11 | 12.74 | 43.68 | 0.32 | 3.12 | 0.54 | 0.79 | 0.01 | 0.33 | 4.79 | 61.09 | 24.18 | 14.73 | sandy loam |
| 30 RS | VS | 5.94 | 5.59 | 5.22 | 9.00 | 0.33 | 15.61 | 127.07 | 6.58 | 13.96 | 2.53 | 0.96 | 0.01 | 0.33 | 17.79 | 56.97 | 31.71 | 11.32 | sandy loam |
| 31 RS | VS | 5.64 | 4.88 | 3.06 | 5.28 | 0.22 | 14.01 | 75.12 | 0.53 | 4.29 | 1.12 | 0.79 | 0.01 | 0.33 | 6.54 | 59.69 | 27.55 | 12.76 | sandy loam |
| 32 R | VS | 7.03 | 5.97 | 1.27 | 2.19 | 0.10 | 12.37 | 26.63 | 2.46 | 9.68 | 0.77 | 0.77 | 0.04 | 0.00 | 11.26 | 65.68 | 20.50 | 13.82 | sandy loam |
| 33 RS | VS | 6.26 | 4.10 | 3.59 | 6.19 | 0.25 | 14.44 | 37.39 | 0.61 | 6.40 | 1.21 | 0.78 | 0.04 | 0.00 | 8.43 | 68.25 | 17.84 | 13.91 | sandy loam |
| 34 R | VR | 6.29 | 4.28 | 1.05 | 1.81 | 0.08 | 12.65 | 6.72 | 0.29 | 0.20 | 0.32 | 0.43 | 0.05 | 0.32 | 1.31 | 64.53 | 20.51 | 14.96 | sandy loam |
| 35 V | VS | 5.96 | 3.39 | 1.09 | 1.88 | 0.11 | 9.59 | 22.85 | 0.02 | 0.21 | 0.45 | 0.80 | 0.04 | 0.48 | 1.98 | 61.45 | 24.60 | 13.95 | sandy loam |
| 36 R | VR | 6.10 | 3.57 | 1.10 | 1.90 | 0.09 | 12.51 | 20.50 | 0.92 | 0.22 | 0.69 | 0.71 | 0.05 | 0.92 | 2.59 | 70.15 | 16.20 | 13.65 | sandy loam |
| 37 R | VR | 4.85 | 3.79 | 1.45 | 2.50 | 0.13 | 11.23 | 18.69 | 0.04 | 0.08 | 0.22 | 0.54 | 0.03 | 1.25 | 2.12 | 47.95 | 37.67 | 14.37 | loam |
| 38 R | VS | 4.91 | 4.31 | 1.85 | 3.19 | 0.15 | 12.37 | 63.78 | 0.22 | 2.52 | 0.42 | 0.86 | 0.01 | 0.52 | 4.33 | 61.09 | 24.18 | 14.73 | sandy loam |
| 39 V | VS | 6.11 | 5.85 | 3.61 | 6.22 | 0.37 | 9.80 | 62.94 | 1.34 | 18.88 | 0.65 | 2.58 | 0.12 | 0.00 | 22.23 | 42.22 | 35.88 | 21.89 | clay loam |
| 40 V | VS | 6.01 | 3.89 | 2.06 | 3.55 | 0.09 | 22.81 | 18.72 | 0.05 | 0.19 | 0.88 | 0.62 | 0.02 | 0.20 | 1.92 | 57.14 | 29.42 | 13.44 | sandy clay loam |
| 41 R | VR | 5.00 | 4.00 | 2.20 | 3.79 | 0.16 | 13.63 | 30.21 | 0.42 | 0.22 | 0.33 | 0.56 | 0.03 | 0.61 | 1.75 | 42.78 | 43.17 | 14.05 | loam |
| 42 R | VR | 5.03 | 4.31 | 3.12 | 5.38 | 0.25 | 12.26 | 49.26 | 0.24 | 0.56 | 0.63 | 0.78 | 0.04 | 0.01 | 2.03 | 43.70 | 41.21 | 15.09 | loam |

RB: Rías Baixas; RS: Ribeira Sacra; R: Ribeiro; M: Monterrei; V: Valdeorras. S.a.: Sampling area; C (%): total carbon; OM: Organic matter; DOC: dissolved organic carbon; DIC: dissolved inorganic carbon; N: total nitrogen; eCEC: effective cation exchange capacity; VR: vineyard row; VS: vineyard strain.

The pH$_{H2O}$ values ranged from acidic (4.61; soil 22-Ribeira Sacra) to alkaline (8.02; soil 6-Valdeorras). The pH is one of the chemical properties of the soil that has the most influence on the availability of Cu. Its regulation is influenced by organic matter content, clays, metal oxides, Al$^{3+}$ concentration in the exchange complex, and parent rock composition. In addition, pH affects the solubility and mobility of nutrients in the soil, which determines the degree of nutrient toxicity or deficiency, as well as Cu toxicity [56]. In general, soils with low pH have higher Cu$^{2+}$ solubility, which increases their mobility and bioavailability. This can lead to contamination problems in other adjacent parts of the ecosystem or toxicity in plants and/or other organisms [57,58].

The pH$_{KCl}$ values were lower than pH$_{H2O}$, especially in the acidic soils. These values ranged from 2.94 (soil 5-Monterrei) to 7.32 (soil 6-Valdeorras). Acidic soils tend to have a lower pH$_{KCl}$ than pH$_{H2O}$, which is related to a higher release of Al$^{3+}$ cations that favor soil acidification. This decrease in pH can lead to an increase in the availability of Cu in the soil,

which can have significant implications for plants and their growth since Cu is an essential nutrient but can also become toxic at high concentrations [59].

The presence of organic matter in vineyard soils plays a fundamental role in Cu retention, especially in the top soil layers [60]. When organic matter comes into contact with metals in the soil, chemical reactions occur that lead to the formation of organometallic compounds, which are capable of binding metals, including Cu [13,61]. The amount of organic matter present in the soil is a key indicator of its quality and fertility. A soil is considered to have a low organic matter content when this value is less than 2% [10]. In the context of this study, the soils analyzed showed organic matter values ranging from 1.07% (soil 13-Ribeiro) to 9% (soil 30-Ribeira Sacra). These results show the diversity of the organic matter levels found in the vineyard soils studied, which may influence the Cu retention capacity and, therefore, the availability/toxicity of this metal for the vines. It should be noted that an adequate amount of organic matter in the soil not only favors Cu fixation, but also improves soil structure, increases its water and nutrient retention capacity, and promotes microbiological activity [62]. Therefore, to optimize growing conditions and healthy vine development, it is important to maintain and increase the organic matter content of vineyard soils through appropriate management practices. Additionally, the presence of adequate soil organic matter can reduce the risk of Cu leaching and its potential negative environmental impacts [63]. Concerning the dissolved organic carbon (DOC), it plays a decisive role in overall soil organic matter and Cu retention properties [64]. The analyzed soils showed a wide variability, ranging from 6.72 mg·kg$^{-1}$ (soil 34-Ribeiro) to 127.07 mg·kg$^{-1}$ (soil 29-Ribeiro). In particular, Cu$^{2+}$ can form a complex with DOM, which directly affects the availability of Cu for vines. In addition, the quality of DOM affects the availability of Cu in the soil more than its quantity [65]. On the other hand, dissolved inorganic carbon (DIC) concentrations ranged from 0.01 mg·kg$^{-1}$ (soil 10-Valdeorras) to 11.47 mg·kg$^{-1}$ (soil 6-Valdeorras). Furthermore, the DIC may affect the availability of nutrients in the soil and play a role in the processes of ion exchange and chemical balance in the root environment of vines.

The effective cation exchange capacity (eCEC) is a fundamental measure that reflects the ability of the surface of soil compounds, such as organic matter, oxides, and clays, to retain cations such as Ca$^{2+}$, Mg$^{2+}$, Na$^+$, and K$^+$ on the surface negative charge. In general, soils are considered to have a strong limitation in cation retention when eCEC values are lower than 4 cmol$_{(+)}$kg$^{-1}$, and a slight limitation when values are between 4 and 7 cmol$_{(+)}$kg$^{-1}$. In this study, the eCEC values of the soils studied ranged from 1.20 (soil 28-Ribeira Sacra) to 22.23 (soil 39-Valdeorras), showing a wide variability in cation retention capacity.

In viticulture, the total nitrogen (N) content of the soil plays a fundamental role in the growth and development of the vine. A low N content is when it is <0.1%, while a high N content is >0.3%. The soils analyzed in this study varied between values below 0.1% (soils 4-Monterrei, 12-Ribeiro, 13-Ribeiro, 34-Ribeiro, 36-Ribeiro, and 40-Valdeorras) and above 0.3% in soil 6-Valdeorras. In soils with low N levels, vines may experience difficulties in their development and growth, which may affect their production. On the other hand, in soils with an N concentration higher than 0.3%, vines may show excessive vigor, which may result in lower grape quality [65].

### 3.2. Total and Available Cu Content and Extraction Efficiency ($E_xE_f$) in Vineyard Soils

The total and available Cu contents of the vineyard soils are presented in Table 3. The results show a wide variability among the selected soils, with total Cu concentrations ranging from 16.4 mg·kg$^{-1}$ (soil 1-Monterrei) to 292.9 mg·kg$^{-1}$ (soil 42-Ribeiro), and available Cu ranging from 0.4 mg·kg$^{-1}$ (soil 15-Valdeorras) to 24.0 mg·kg$^{-1}$ (soil 34-Ribeiro). These values indicate that there is considerable heterogeneity in the Cu content of the vineyard soils studied.

**Table 3.** Copper contents (total and available) in vineyard soils (mg·kg$^{-1}$) and extraction efficiency ($E_xE_f$).

| Soil | Sampling Area | Total Cu mg kg$^{-1}$ | Available Cu mg kg$^{-1}$ | $E_xE_f$ (%) | Soil | Sampling Area | Total Cu mg kg$^{-1}$ | Available Cu mg kg$^{-1}$ | $E_xE_f$ (%) |
|---|---|---|---|---|---|---|---|---|---|
| **1** M | VR | 16.4 | 4.2 | 25.8 | **22** RS | VR | 129.4 | 4.3 | 3.3 |
| **2** RB | VS | 33.1 | 3.7 | 11.0 | **23** M | VR | 135.6 | 2.2 | 1.6 |
| **3** RS | VR | 34.2 | 2.0 | 5.7 | **24** RS | VR | 153.2 | 1.2 | 0.8 |
| **4** M | VR | 42.3 | 21.3 | 50.4 | **25** RS | VR | 164.3 | 19.4 | 11.8 |
| **5** M | VS | 44.5 | 2.4 | 5.3 | **26** V | VS | 166.9 | 1.4 | 0.8 |
| **6** V | VR | 49.3 | 3.2 | 6.5 | **27** RS | VS | 170.4 | 1.4 | 0.8 |
| **7** RS | VS | 60.7 | 2.8 | 4.6 | **28** RB | VR | 182.5 | 7.5 | 4.1 |
| **8** RB | VR | 64.3 | 7.5 | 11.6 | **29** R | VR | 187.6 | 8.3 | 4.4 |
| **9** RB | VS | 65.7 | 2.4 | 3.6 | **30** RS | VS | 192.3 | 8.1 | 4.2 |
| **10** V | VR | 66.8 | 23.7 | 35.5 | **31** RS | VS | 200.8 | 7.6 | 3.8 |
| **11** V | VS | 69.5 | 11.1 | 15.9 | **32** R | VS | 201.8 | 1.5 | 0.8 |
| **12** R | VS | 73.6 | 10.0 | 13.6 | **33** RS | VS | 204.0 | 3.4 | 1.6 |
| **13** R | VR | 74.6 | 3.2 | 4.3 | **34** R | VR | 204.0 | 24.0 | 11.8 |
| **14** R | VR | 82.4 | 6.7 | 8.1 | **35** V | VS | 216.0 | 13.7 | 6.4 |
| **15** V | VS | 104.0 | 0.4 | 0.4 | **36** R | VR | 219.4 | 15.2 | 7.0 |
| **16** V | VR | 109.2 | 2.3 | 2.1 | **37** R | VR | 229.8 | 4.2 | 1.8 |
| **17** V | VR | 114.0 | 12.0 | 10.5 | **38** R | VS | 230.8 | 11.5 | 5.0 |
| **18** RB | VS | 117.5 | 2.3 | 1.9 | **39** V | VS | 248.8 | 15.1 | 6.1 |
| **19** RS | VS | 117.7 | 7.8 | 6.7 | **40** V | VS | 268.4 | 18.7 | 7.0 |
| **20** R | VR | 122.2 | 18.2 | 14.9 | **41** R | VR | 285.5 | 7.6 | 2.7 |
| **21** V | VS | 126.7 | 3.1 | 2.5 | **42** R | VR | 292.9 | 8.0 | 2.7 |

RB: Rías Baixas; RS: Ribeira Sacra; R: Ribeiro; M: Monterrei; V: Valdeorras. VR: vineyard row; VS: vineyard strain.

Concerning the generic Cu reference limits for Galician soils [13], it should be noted that many of the vineyard soils analyzed in this study exceed the maximum content recommended for ecosystem protection, which is 50 mg kg$^{-1}$ (soils from 7 to 42). In addition, many of these vineyard soils also exceed the phytotoxic limit of 100 mg kg$^{-1}$ (soils from 15 to 42) [12].

The high Cu content in most of the soils studied is closely related to the accumulation of this metal as a result of the phytosanitary treatments carried out on the vines over the years with copper-based fungicides [66]. These treatments, although necessary to control fungal diseases and maintain the development and growth of the vines, can cause a high increase in the Cu content of the soil. It is therefore essential to take these reference levels into account when adopting pest and disease management measures with Cu-based fungicides, as exceeding them can have negative consequences for parts of the adjacent ecosystems as well as for the vineyards. Indeed, high levels of Cu in the soil can promote its leaching and affect the quality of groundwater and surface water [67]. In addition, an excess of Cu in the soil can be phytotoxic to other plants or crops in the vicinity of the vineyard [12].

The available Cu content of soils is much lower than the total Cu content. Available Cu levels < 1 mg kg$^{-1}$ are considered deficient, while available levels > 25 mg kg$^{-1}$ may be toxic to grapevines, especially in acidic soils [10] (extracted used: 0.005 M DTPA, 0.01 M CaCl$_2$, and 0.1 M TEA—pH 7.3—ratio 1 g soil /10 mL solution). None of the soils exceeded 25 mg kg$^{-1}$, while 15 soils were below 1 mg kg$^{-1}$ of available Cu. In addition, the extraction efficiency ($E_xE_f$) of Cu in each vineyard soil was calculated to show the percentage of Cu released [53] (Table 3).

Pearson's bivariate correlation analysis for data of soils sampled in vineyard strains (VS) and vineyard rows (VR) indicates that there is no positive correlation between total and available Cu contents in the soils (Table 4). In fact, many of the soils with high Cu contents show low available Cu contents and therefore low $E_xE_f$ (e.g., soils 26-Valdeorras, 27-Ribeiro Sacra, 32-Ribeiro and 37-Ribeiro), while soils with low total Cu contents show high available Cu contents and high $E_xE_f$ (e.g., soils 4 and 10). It was found that there is a

positive correlation between available Cu content and $E_xE_f$ for soils sampled in vineyard strains (VS) (R = 0.63) and for soils sampled in vineyard rows (VR) (R = 0.62). Correlation analysis also indicates that there is no specific correlation between total or available Cu content and soil characteristics for vineyard soil samples. This suggests that it may be a combination of factors that influence Cu availability or total content in vineyard soils.

**Table 4.** Pearson's bivariate correlation analysis (R values) for data of soils sampled in vineyard strains (VS) and vineyard rows (VR) and data from assays with *L. perenne*.

| | pH$_{H2O}$ | pH$_{KCl}$ | C(%) | OM | N | C/N | DOC | DIC | Ca$^{2+}$ | Mg$^{2+}$ | K$^+$ | Na$^+$ | Al$^{3+}$ | eCEC | Sand | Silt | Clay | Total Cu | Avail. Cu | $E_xE_f$ | Bm | Cu a.p. | BF |
|---|---|---|---|---|---|---|---|---|---|---|---|---|---|---|---|---|---|---|---|---|---|---|---|
| **SAMPLING AREA: VS** | | | | | | | | | | | | | | | | | | | | | | | |
| Total Cu | −0.13 | 0.02 | −0.06 | −0.06 | −0.03 | −0.03 | 0.24 | −0.06 | 0.04 | −0.01 | 0.34 | −0.27 | −0.29 | 0.03 | 0.00 | 0.03 | −0.08 | - | - | - | - | - | - |
| Avail. Cu | −0.19 | −0.27 | −0.22 | −0.22 | −0.28 | 0.14 | −0.08 | −0.26 | −0.25 | −0.09 | 0.06 | −0.09 | −0.10 | −0.26 | 0.02 | 0.04 | −0.19 | 0.25 | - | - | - | - | - |
| $E_xE_f$ | −0.08 | −0.26 | −0.15 | −0.15 | −0.22 | 0.11 | −0.23 | −0.18 | −0.23 | −0.10 | −0.13 | 0.04 | 0.10 | −0.06 | 0.12 | −0.14 | −0.41 | 0.63 | - | - | - | - | - |
| Bm | 0.26 | 0.35 | 0.50 | 0.50 | 0.56 | −0.01 | 0.20 | 0.30 | 0.53 | 0.27 | 0.36 | 0.03 | −0.25 | 0.54 | 0.03 | −0.02 | −0.02 | 0.13 | −0.31 | −0.30 | - | - | - |
| Cu a.p. | −0.17 | −0.30 | −0.25 | −0.25 | −0.28 | −0.11 | −0.20 | −0.23 | −0.24 | −0.19 | −0.09 | 0.13 | 0.26 | −0.23 | −0.05 | 0.06 | 0.00 | 0.18 | 0.31 | 0.29 | −0.17 | - | - |
| BF | 0.10 | 0.04 | −0.07 | −0.07 | 0.02 | −0.23 | −0.13 | 0.03 | 0.02 | −0.11 | −0.02 | −0.02 | 0.08 | 0.01 | −0.09 | −0.06 | 0.53 | −0.10 | −0.55 | −0.34 | 0.05 | −0.08 | - |
| **SAMPLING AREA: VR** | | | | | | | | | | | | | | | | | | | | | | | |
| Total Cu | −0.23 | −0.03 | −0.01 | −0.01 | 0.02 | −0.04 | 0.23 | −0.10 | −0.04 | −0.08 | 0.28 | −0.27 | −0.29 | −0.06 | −0.11 | 0.16 | −0.10 | - | - | - | - | - | - |
| Avail. Cu | −0.18 | −0.27 | −0.22 | −0.22 | −0.27 | 0.14 | −0.07 | −0.25 | −0.24 | −0.09 | 0.06 | −0.09 | −0.10 | −0.26 | 0.02 | 0.04 | −0.19 | 0.22 | - | - | - | - | - |
| $E_xE_f$ | −0.04 | −0.24 | −0.16 | −0.16 | −0.22 | 0.11 | −0.23 | −0.16 | −0.21 | −0.08 | −0.12 | 0.04 | 0.11 | −0.20 | −0.03 | 0.07 | −0.13 | −0.42 | 0.62 | - | - | - | - |
| Bm | 0.21 | 0.33 | 0.51 | 0.51 | 0.57 | −0.02 | 0.21 | 0.27 | 0.48 | 0.25 | 0.36 | 0.02 | −0.28 | 0.49 | 0.00 | 0.01 | −0.02 | 0.16 | −0.30 | −0.30 | - | - | - |
| Cu a.p. | −0.13 | −0.28 | −0.26 | −0.26 | −0.29 | −0.11 | −0.20 | −0.22 | −0.22 | −0.17 | −0.09 | 0.14 | 0.26 | −0.20 | −0.02 | 0.02 | 0.00 | 0.11 | 0.31 | 0.30 | −0.18 | - | - |
| BF | 0.13 | 0.05 | −0.08 | −0.08 | 0.01 | −0.22 | −0.13 | 0.04 | 0.04 | −0.09 | −0.01 | −0.01 | 0.08 | 0.03 | −0.06 | −0.09 | 0.53 | −0.13 | −0.55 | −0.32 | 0.03 | −0.07 | - |

C (%): total carbon; OM: Organic matter; DOC: dissolved organic carbon; DIC: dissolved inorganic carbon; N: total nitrogen; eCEC: effective cation exchange capacity; extraction efficiency ($E_xE_f$); Biomass (Bm); Cu in aerial part of *L. perenne* (Cu a.p.); Bioaccumulation factor (BF). Color intensity: Red indicates a value of R = 1, blue indicates a value of R = −1, white R = 0.

A bivariate Pearson's correlation analysis was also carried out, separating the soils into those with total Cu contents below the limit considered phytotoxic (<100 mg kg$^{-1}$) and those with contents above 100 mg kg$^{-1}$ (above the phytotoxic limit), regardless of the sampling zone (VS or VR) (Table 5). The rest of the results of the correlation analysis were similar to those obtained when the soils were separated by sampling zones. No specific correlation was observed between total and available Cu contents and a strong correlation between available Cu content and $E_xE_f$ was observed, (R = 0.87 and R = 0.90), while there is also no specific correlation between total or available Cu content and soil characteristics.

Principal Component Analysis (PCA) was carried out to determine which variables could explain the different types of vineyard soils studied and their possible relationship with Cu contents. The results of the PCA are shown in Figures 1 and 2 and Tables 5 and 6. The PCA analysis of the data from the trials with soils sampled in the corridor zone (P) (Figure 3 and Table 6) allowed us to reduce the initial dimension of the dataset to two dimensions that explain 70.85% of the variance of the data (Component **1**: 42.25% and Component **2**: 28.71%) (Table 6). In the case of the soils sampled in the stock zone (C) (Figure 1 and Table 6), PCA also allowed us to reduce the initial dimension of the dataset to two dimensions that explain 76.51% of the variance of the data (Component **1**: 49.95% and Component **2**: 26.57%) (Table 6).

The available Cu content, as well as the $E_xE_f$ and its relationship with the soil characteristics, in soils sampled in both VS and VR, can be explained by component **2**. A negative relationship between available Cu content and $E_xE_f$, organic matter content, DOC, and nitrogen content is observed. This means that in soils with low organic matter and nitrogen content, the availability of Cu in the soil can increase. This may be due to the fact that in soils poor in organic matter and nitrogen, there is less competition for adsorption sites and therefore less Cu retention, which facilitates its release and availability (Figures 1 and 2).

**Table 5.** Pearson's bivariate correlation analysis (R values) for data of soils with total Cu contents < 100 mg kg⁻¹ (A) (below phytotoxic level) and > 100 mg kg⁻¹ (B) (above phytotoxic level) and data from assays with *L. perenne*.

| | $pH_{H2O}$ | $pH_{KCl}$ | C(%) | OM | N | C/N | DOC | DIC | $Ca^{2+}$ | $Mg^{2+}$ | $K^+$ | $Na^+$ | $Al^{3+}$ | eCEC | Sand | Silt | Clay | Total Cu | Avail. Cu | $E_xE_f$ | Bm | Cu a.p. | BF |
|---|---|---|---|---|---|---|---|---|---|---|---|---|---|---|---|---|---|---|---|---|---|---|---|
| **Soils with total Cu contents below phytotoxic level (<100 mg kg⁻¹)** | | | | | | | | | | | | | | | | | | | | | | | |
| Total Cu | −0.11 | 0.07 | −0.26 | −0.26 | −0.35 | 0.19 | −0.27 | −0.05 | −0.09 | −0.01 | 0.14 | 0.41 | −0.01 | −0.09 | 0.33 | −0.32 | −0.31 | | 0.19 | −0.24 | −0.09 | −0.30 | −0.44 |
| Avail. Cu | −0.05 | −0.18 | −0.32 | −0.33 | −0.43 | 0.10 | −0.47 | −0.16 | −0.25 | −0.15 | 0.16 | −0.10 | −0.16 | −0.29 | −0.19 | 0.25 | −0.24 | 0.19 | | 0.87 | −0.65 | 0.11 | −0.78 |
| $E_xE_f$ | −0.05 | −0.23 | −0.25 | −0.25 | −0.30 | 0.00 | −0.44 | −0.17 | −0.27 | −0.18 | −0.02 | −0.21 | −0.08 | −0.31 | −0.27 | 0.32 | −0.13 | −0.24 | 0.87 | | −0.62 | 0.37 | −0.53 |
| Bm | 0.40 | 0.41 | 0.58 | 0.58 | 0.62 | 0.11 | 0.45 | 0.44 | 0.50 | 0.34 | 0.04 | −0.07 | −0.14 | 0.52 | 0.03 | −0.09 | 0.31 | −0.09 | −0.65 | −0.62 | | −0.52 | 0.36 |
| Cu a.p. | −0.31 | −0.22 | −0.22 | −0.22 | −0.26 | −0.11 | −0.29 | −0.29 | −0.16 | 0.10 | −0.35 | 0.45 | 0.40 | −0.10 | −0.03 | 0.04 | −0.01 | −0.30 | 0.11 | 0.37 | −0.52 | | 0.16 |
| BF | −0.21 | −0.20 | 0.19 | 0.19 | 0.30 | −0.25 | 0.14 | −0.16 | −0.01 | 0.04 | 0.08 | 0.05 | 0.30 | 0.04 | 0.03 | −0.07 | 0.21 | −0.44 | −0.78 | −0.53 | 0.36 | 0.16 | |
| **Soils with total Cu contents above phytotoxic level (>100 mg kg⁻¹)** | | | | | | | | | | | | | | | | | | | | | | | |
| Total Cu | −0.40 | −0.31 | 0.05 | 0.05 | 0.02 | 0.22 | −0.13 | −0.25 | −0.14 | −0.33 | 0.20 | 0.09 | −0.06 | −0.15 | −0.04 | 0.15 | −0.26 | | 0.34 | −0.02 | 0.24 | 0.17 | −0.40 |
| Avail. Cu | −0.27 | −0.36 | −0.15 | −0.15 | −0.18 | 0.20 | −0.08 | −0.34 | −0.25 | −0.09 | 0.03 | −0.07 | −0.01 | −0.24 | 0.18 | −0.12 | −0.20 | 0.34 | | 0.90 | −0.19 | 0.08 | −0.59 |
| $E_xE_f$ | −0.19 | −0.24 | −0.14 | −0.14 | −0.15 | 0.11 | 0.01 | −0.27 | −0.20 | 0.12 | −0.05 | −0.14 | −0.03 | −0.18 | 0.17 | −0.12 | −0.18 | −0.02 | 0.90 | | −0.24 | −0.04 | −0.54 |
| Bm | 0.15 | 0.33 | 0.51 | 0.51 | 0.59 | −0.08 | 0.21 | 0.22 | 0.50 | 0.23 | 0.41 | 0.31 | −0.48 | 0.50 | 0.00 | 0.04 | −0.08 | 0.24 | −0.19 | −0.24 | | −0.02 | −0.01 |
| Cu a.p. | 0.00 | −0.19 | −0.19 | −0.19 | −0.21 | −0.17 | −0.13 | 0.07 | −0.06 | −0.18 | −0.14 | 0.01 | 0.13 | −0.08 | 0.25 | −0.27 | −0.01 | 0.17 | 0.08 | −0.04 | −0.02 | | 0.04 |
| BF | 0.23 | 0.10 | −0.14 | −0.14 | −0.05 | −0.26 | −0.21 | 0.10 | 0.04 | −0.13 | −0.05 | 0.15 | 0.17 | 0.03 | −0.06 | −0.15 | 0.56 | −0.40 | −0.59 | −0.54 | −0.01 | 0.04 | |

C (%): total carbon; OM: Organic matter; DOC: dissolved organic carbon; DIC: dissolved inorganic carbon; N: total nitrogen; eCEC: effective cation exchange capacity; extraction efficiency ($E_xE_f$); Biomass (Bm); Cu in aerial part of *L. perenne* (Cu a.p.); Bioaccumulation factor (BF). Color intensity: Red indicates a value of R = 1, blue indicates a value of R = −1, white R = 0.

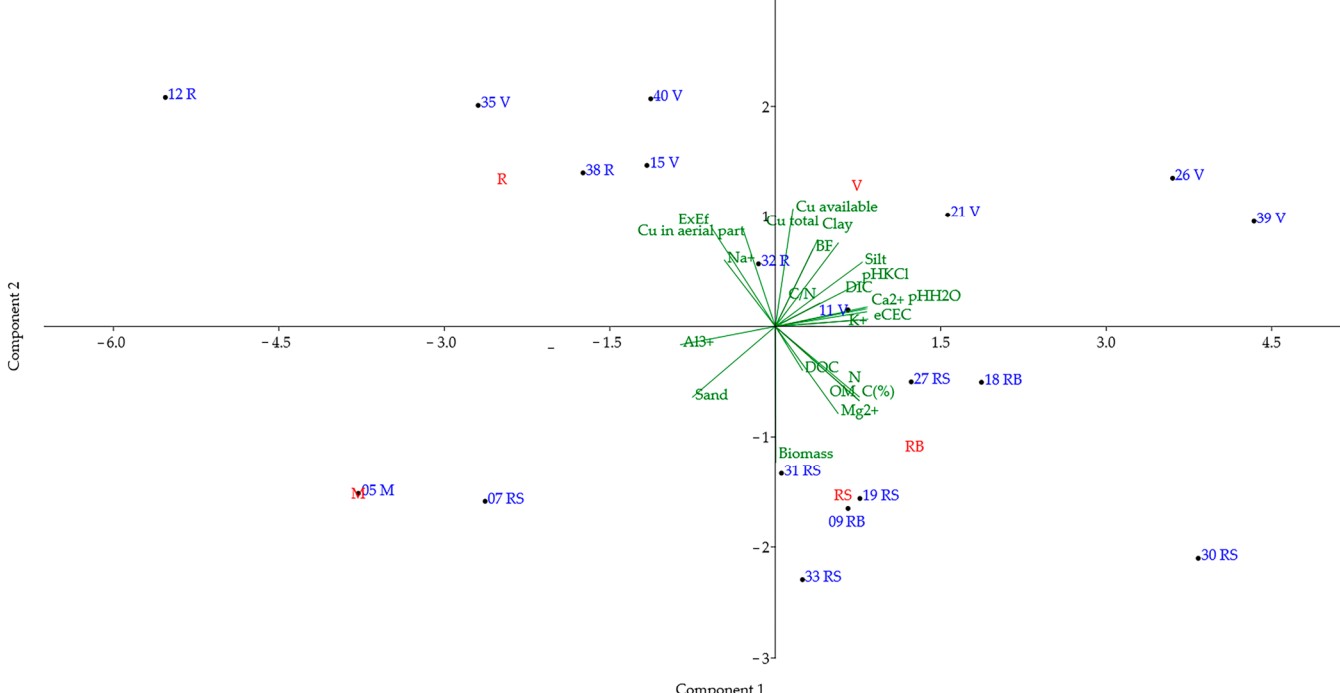

**Figure 1.** Scatter plot of principal component analysis (PCA) for soil data sampled in vineyard strains (VS) and data from assays with *L. perenne*.

### 3.3. Biomass of L. perenne, Copper Content, and the Bioaccumulation Factor

Table 7 shows the data related to the dry biomass obtained from the aerial part of *L. perenne*, as well as the Cu content in the aerial part of *L. perenne* obtained in the pot experiments. From the results of the dry biomass weights, it was found that the biomass content of *L. perenne* did not seem to be affected by the total Cu content in the soil. In fact, *L. perenne* biomass in soils with Cu concentrations above the phytotoxic limit (100 mg kg⁻¹) did not differ from soils with Cu values < 100 mg kg⁻¹ (Figure 2). Furthermore, correlation analysis showed that *L. perenne* biomass for soils sampled in VR was related to organic matter content (R = 0.50), total N (R = 0.56), and eCEC (R = 0.54), with $Ca^{2+}$ (R = 0.53) being

the cation of the exchange complex that contributed most to *L. perenne* biomass content. Similar results were found for soils sampled in VS, where correlation analysis indicated that *L. perenne* biomass was also related to organic matter (R = 0.51), total N (R = 0.57), eCEC (R = 0.49), and $Ca^{2+}$ (R = 0.48). *Lolium perenne* is known for its ability to respond positively to soils with high N, organic matter, and cation exchange capacity, which affects its growth and development [68].

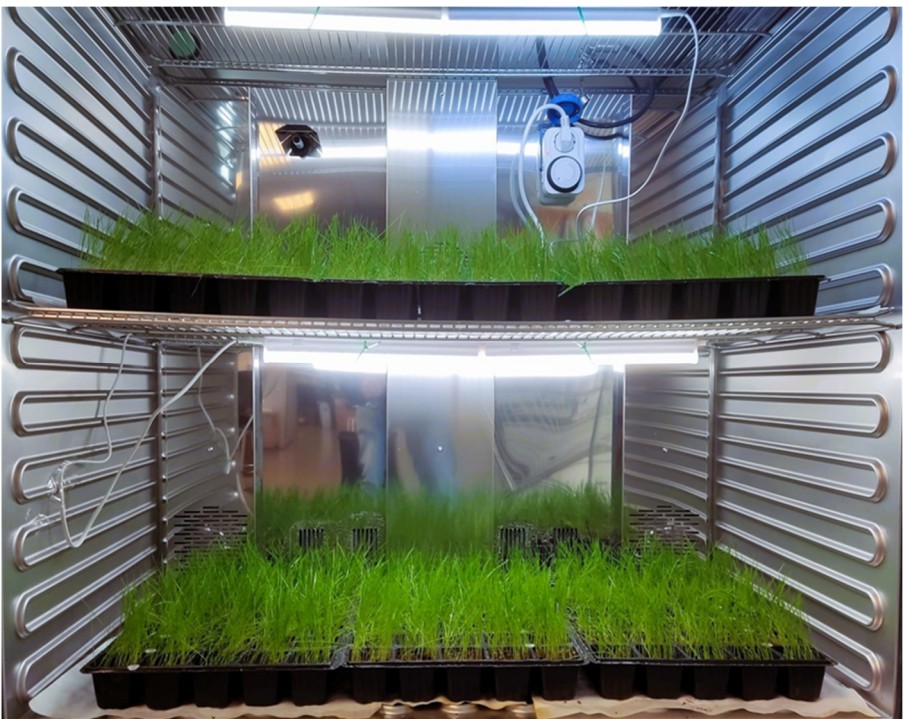

**Figure 2.** Image of the pot experiment with vineyard soils and *L. perenne* in a climatic chamber after one month of growth.

**Table 6.** Eigenvalues, percent variance explained, and loadings of each PCA component for data from soils sampled in vineyard rows (VR) and vineyard strains (VS) and data from assays with *L. perenne* (linked to Figures 1 and 2).

| Soils Sampled in Vineyard Rows (VR) | | | |
|---|---|---|---|
| **PC** | **Eigenvalues** | | **% Variance** |
| **1** | 9.694 | | 42.148 |
| **2** | 6.602 | | 28.704 |
| **3** | 4.591 | | 19.961 |
| **4** | 2.113 | | 9.187 |
| | **PC 1** | **PC 2** | **PC 3** | **PC 4** |

| | **PC 1** | **PC 2** | **PC 3** | **PC 4** |
|---|---|---|---|---|
| **01 M** | −0.996 | −1.236 | 1.575 | −2.04 |
| **03 RS** | −1.088 | 0.709 | 0.088 | −1.138 |
| **04 M** | −1.88 | −2.599 | 0.503 | −1.008 |
| **06 V** | 6.897 | 0.597 | 3.817 | −0.17 |
| **08 RB** | 3.803 | 2.62 | −0.558 | −2.253 |
| **10 V** | −0.162 | −2.797 | −0.864 | −0.111 |
| **13 R** | −2.284 | −0.959 | −1.835 | −0.988 |
| **14 R** | −1.133 | −0.439 | −0.843 | −0.569 |
| **16 V** | 0.444 | −1.362 | 1.669 | 0.488 |
| **17 V** | 2.018 | −0.574 | 0.966 | 0.639 |
| **20 R** | 0.929 | 1.467 | 0.339 | 1.837 |

**Table 6.** *Cont.*

| | Soils Sampled in Vineyard Rows (VR) | | | |
|---|---|---|---|---|
| PC | Eigenvalues | | % Variance | |
| 22 RS | −0.869 | 4.436 | 2.025 | 0.986 |
| 23 M | −0.002 | 0.016 | −0.086 | −0.797 |
| 24 RS | −0.429 | 3.015 | 1.459 | −1.004 |
| 25 RS | −0.965 | 0.02 | −0.749 | 0.356 |
| 28 RB | 1.04 | −1.798 | −1.276 | 0.29 |
| 29 R | −0.385 | 0.525 | −1.048 | 1.197 |
| 34 R | −1.203 | −2.035 | −1.472 | 0.632 |
| 36 R | −2.298 | −1.131 | −1.351 | 1.14 |
| 37 R | −1.597 | −1.095 | −0.764 | 0.521 |
| 41 R | −0.681 | 0.199 | −0.723 | 0.715 |
| 42 R | 0.841 | 2.421 | −0.872 | 1.276 |
| | Soils sampled in vineyard strains (VS) | | | |
| PC | Eigenvalues | | % variance | |
| 1 | 11.487 | | 49.945 | |
| 2 | 6.112 | | 26.572 | |
| 3 | 2.988 | | 12.992 | |
| 4 | 2.413 | | 10.491 | |
| | PC 1 | PC 2 | PC 3 | PC 4 |
| 05 M | −3.78 | −1.518 | −0.729 | −1.402 |
| 07 RS | −2.631 | −1.589 | −0.893 | −0.893 |
| 09 RB | 0.659 | −1.654 | −1.645 | 0.38 |
| 11 V | 0.658 | 0.151 | 0.728 | −1.347 |
| 12 R | −5.53 | 2.082 | 0.442 | 0.96 |
| 15 V | −1.164 | 1.466 | −3.879 | −1.874 |
| 18 RB | 1.869 | −0.505 | −1.437 | 0.937 |
| 19 RS | 0.767 | −1.564 | 2.058 | −0.358 |
| 21 V | 1.563 | 1.015 | −1.037 | 0.877 |
| 26 V | 3.602 | 1.35 | −0.988 | 0.577 |
| 27 RS | 1.232 | −0.501 | −0.16 | 0.633 |
| 30 RS | 3.834 | −2.102 | 3.193 | 1.38 |
| 31 RS | 0.055 | −1.336 | 1.092 | −0.187 |
| 32 R | −0.153 | 0.567 | −0.544 | 1.561 |
| 33 RS | 0.246 | −2.295 | −0.033 | 0.28 |
| 35 V | −2.694 | 2.01 | 0.576 | −0.04 |
| 38 R | −1.743 | 1.399 | 1.421 | −0.157 |
| 39 V | 4.34 | 0.954 | 0.34 | 0.652 |
| 40 V | −1.13 | 2.069 | 1.494 | −1.979 |

Regarding the ability of *L. perenne* to grow in vineyard soils with high Cu contents, it is important to consider that this species is considered tolerant to the presence of Cu in the soil [69]. The results of Cu in the aerial part and the bioaccumulation factor (the ratio between the Cu content in the aerial part of *L. perenne* and the available Cu content in the soil) show that the Cu contents accumulated in the aerial part of the plant are low, with percentages in the aerial part that, in most cases, are less than 1% of the available Cu content in the soil (Table 4). Only the soils 15-Valdeorras, 24-Ribeira Sacra, 26-Valdeorras, and 32-Ribeiro have a BF ≥ 1%. Correlation analysis showed that BF is negatively correlated with available Cu content, both in the soils sampled in VR (R = 0.55) and VS (R = 0.55). Thus, there is an inverse relationship between available Cu content in vineyard soils and Cu accumulation in the aerial part of *L. perenne*. In other words, Cu accumulation in the aerial part of *L. perenne* may decrease when there is higher Cu availability in the soil, suggesting that there may be defense mechanisms in *L. perenne* that prevent Cu accumulation in its aerial part [70].

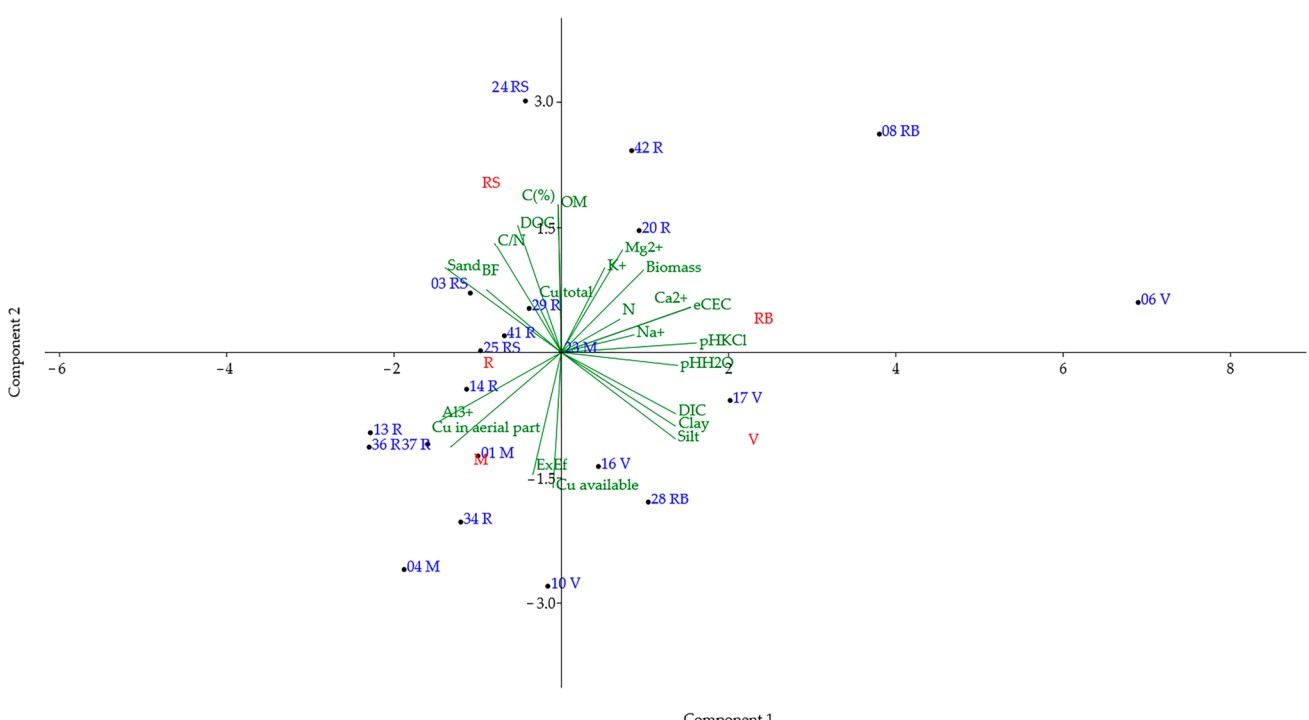

**Figure 3.** Scatter plot of principal component analysis (PCA) for soil data sampled in vineyard rows (VR) and data from assays with *L. perenne*.

**Table 7.** Dry biomass production (mg kg$^{-1}$) of *L. perenne*, Cu content in the aerial part, and bioaccumulation factor (BF).

| Soil | Sampling Area | Biomass | Cu in Aerial Part | BF | Soil | Sampling Area | Biomass | Cu in Aerial Part | BF |
|------|------|------|------|------|------|------|------|------|------|
|  |  | mg kg$^{-1}$ | mg kg$^{-1}$ | (%) |  |  | mg kg$^{-1}$ | mg kg$^{-1}$ | (%) |
| **1** M | VR | 0.769 ± 0.048 | 0.023 ± 0.002 | 0.55 | **22** RS | VR | 0.690 ± 0.004 | 0.011 ± 0.001 | 0.26 |
| **2** RB | VS | 0.731 ± 0.030 | 0.015 ± 0.001 | 0.41 | **23** M | VR | 0.679 ± 0.036 | 0.013 ± 0.001 | 0.59 |
| **3** RS | VR | 0.744 ± 0.010 | 0.009 ± 0.001 | 0.47 | **24** RS | VR | 0.831 ± 0.127 | 0.015 ± 0.001 | 1.27 |
| **4** M | VR | 0.621 ± 0.021 | 0.017 ± 0.004 | 0.08 | **25** RS | VR | 0.724 ± 0.099 | 0.013 ± 0.001 | 0.07 |
| **5** M | VS | 0.858 ± 0.078 | 0.011 ± 0.008 | 0.46 | **26** V | VS | 0.837 ± 0.017 | 0.015 ± 0.001 | 1.05 |
| **6** V | VR | 0.847 ± 0.048 | 0.009 ± 0.003 | 0.27 | **27** RS | VS | 0.859 ± 0.103 | 0.012 ± 0.001 | 0.85 |
| **7** RS | VS | 0.83 ± 0.103 | 0.008 ± 0.002 | 0.30 | **28** RB | VR | 0.778 ± 0.099 | 0.014 ± 0.001 | 0.18 |
| **8** RB | VR | 0.883 ± 0.077 | 0.013 ± 0.005 | 0.17 | **29** R | VR | 0.699 ± 0.103 | 0.012 ± 0.001 | 0.15 |
| **9** RB | VS | 0.847 ± 0.016 | 0.013 ± 0.003 | 0.55 | **30** RS | VS | 0.99 ± 0.166 | 0.015 ± 0.004 | 0.18 |
| **10** V | VR | 0.643 ± 0.042 | 0.014 ± 0.002 | 0.06 | **31** RS | VS | 0.979 ± 0.042 | 0.013 ± 0.001 | 0.17 |
| **11** V | VS | 0.788 ± 0.006 | 0.014 ± 0.001 | 0.12 | **32** R | VS | 0.922 ± 0.011 | 0.015 ± 0.001 | 1.00 |
| **12** R | VS | 0.695 ± 0.031 | 0.017 ± 0.001 | 0.17 | **33** RS | VS | 1.061 ± 0.053 | 0.012 ± 0.001 | 0.37 |
| **13** R | VR | 0.685 ± 0.053 | 0.016 ± 0.001 | 0.51 | **34** R | VR | 0.765 ± 0.019 | 0.019 ± 0.001 | 0.08 |
| **14** R | VR | 0.691 ± 0.012 | 0.013 ± 0.001 | 0.19 | **35** V | VS | 0.643 ± 0.046 | 0.019 ± 0.001 | 0.14 |
| **15** V | VS | 0.672 ± 0.052 | 0.012 ± 0.001 | 2.98 | **36** R | VR | 0.708 ± 0.037 | 0.025 ± 0.008 | 0.16 |
| **16** V | VR | 0.654 ± 0.019 | 0.016 ± 0.002 | 0.70 | **37** R | VR | 0.727 ± 0.073 | 0.016 ± 0.002 | 0.37 |
| **17** V | VR | 0.746 ± 0.027 | 0.013 ± 0.002 | 0.10 | **38** R | VS | 0.645 ± 0.057 | 0.017 ± 0.001 | 0.15 |
| **18** RB | VS | 0.855 ± 0.006 | 0.014 ± 0.001 | 0.60 | **39** V | VS | 0.957 ± 0.047 | 0.013 ± 0.003 | 0.09 |
| **19** RS | VS | 0.605 ± 0.227 | 0.012 ± 0.001 | 0.15 | **40** V | VS | 0.569 ± 0.038 | 0.011 ± 0.001 | 0.06 |
| **20** R | VR | 0.732 ± 0.009 | 0.014 ± 0.001 | 0.07 | **41** R | VR | 0.689 ± 0.087 | 0.013 ± 0.001 | 0.17 |
| **21** V | VS | 0.613 ± 0.049 | 0.012 ± 0.001 | 0.40 | **42** R | VR | 0.961 ± 0.002 | 0.012 ± 0.001 | 0.14 |

RB: Rías Baixas; RS: Ribeira Sacra; R: Ribeiro; M: Monterrei; V: Valdeorras. VR: vineyard row; VS: vineyard strain.

The exclusion of Cu in the aerial part of *L. perenne* may be beneficial in reducing Cu transfer through the food chain. This makes this species a good choice as a ground

cover without increasing the risk of Cu transfer to other organisms. Although these results suggest that *L. perenne* is not suitable for Cu phytoextraction in vineyard soils, it can play a valuable role as a cover crop due to its ability to grow in high Cu environments and improve edaphic soil conditions [69,71]. In addition, its use as a cover crop can have other benefits, such as improving soil structure by increasing porosity and water-holding capacity and preventing soil erosion.

The PCA analysis with the data of the assays with soils sampled in VR (Figure 3 and Table 6) allowed us to explain the effect of available Cu on the biomass of *L. perenne* and the Cu content in the aerial part of the plant. Component **1** for soils sampled in VR explained the negative relationship between *L. perenne* biomass and aerial Cu content, while component **2** explained the positive relationship between aerial Cu content and available Cu content. This means that a higher Cu content in the aerial part of *L. perenne* is associated with high available Cu contents in the soils sampled in VR. Furthermore, a higher Cu content in the aerial part has a negative effect on the biomass of *L. perenne*, a parameter that, as mentioned above, is not affected by the total Cu content in the soil (Figure 3 and Table 3). In line with these results, component **1** also explains a positive relationship between *L. perenne* biomass and soil characteristics such as eCEC and exchange complex cations such as $Ca^{2+}$, $Mg^{2+}$, and $K^+$, as well as with soil N content. In addition, $Al^{3+}$ levels may be detrimental to the growth of *L. perenne*.

The PCA analysis with the data of the assays with soils sampled in VS (Figure 1 and Table 7) also served to explain the effect of available Cu on *L. perenne* biomass and Cu content in the aerial part of the plant. In particular, component **2** explains most of the results, where it has been observed that the biomass of *L. perenne* is negatively related to the Cu content in the aerial part of the plant and to the Cu available in the soil. In addition, the growth of *L. perenne* is mainly favored by the content of organic matter and N.

## 4. Conclusions

The vineyard soils of the different D.O.s of Galicia show great variability in their physico-chemical characteristics and significant variations within each D.O., including differences in texture, pH, organic matter content, cation exchange capacity, and nitrogen content. The total and available Cu contents of the vineyard soils studied varied widely. Many of the soils analyzed exceeded the reference limits established for ecosystem protection and phytotoxicity, indicating that phytosanitary treatments with Cu-based fungicides used in vineyards have led to significant Cu accumulation in most of the soils. The total Cu content of the soil does not seem to affect the biomass of *L. perenne*. Even in soils with Cu concentrations above the phytotoxic limit, no differences in plant biomass were observed compared to soils with lower Cu levels. Plant biomass is related to soil characteristics such as organic matter content, N, and cation exchange capacity, indicating that these factors are more determinant for plant growth. Although *L. perenne* is considered to be tolerant to the presence of Cu in the soil, an inverse relationship was found between available Cu content in the soil and Cu accumulation in the aerial part of the plant. Therefore, although *L. perenne* is not suitable for phytoextraction of Cu from vineyard soils, it may play a valuable role as a phytoremediation species and as a ground cover in vineyard soils, helping to improve soil conditions and prevent mobility and transfer of Cu to other parts of the ecosystem or through the food chain. This study provides the basis for future scientific research aimed at achieving sustainable agriculture and developing techniques for the reclamation of vineyard soils affected by high Cu concentrations through the use of plant cover based on the use of *L. perenne*.

**Author Contributions:** Conceptualization, D.F.-C. and D.A.-L.; methodology, D.F.-C. and M.A.-E.; software, D.A.-L.; validation, R.V.-B., M.A.-E., D.F.-C. and D.A.-L.; formal analysis, R.V.-B.; investigation, R.V.-B., M.A.-E., D.F.-C. and D.A.-L.; resources, M.A.-E. and D.F.-C.; data curation, R.V.-B.; writing—original draft preparation, R.V.-B.; writing—review and editing, M.A.-E., D.F.-C. and D.A.-L.; visualization, M.A.-E. and D.A.-L.; supervision, D.F.-C. and D.A.-L.; project administration, D.F.-C.; funding acquisition, D.F.-C. All authors have read and agreed to the published version of the manuscript.

**Funding:** This work was funded by the COPPEREPLACE project, which has received funding from (grant agreement SOE4/P1/E1000) FEDER funds through the INTERREG- SUDOE program, and by Xunta de Galicia via the BV1 research group (ED431C 2017/62-GRC).

**Data Availability Statement:** All data are shown in the manuscript.

**Acknowledgments:** Daniel Arenas-Lago thanks the Ministerio de Ciencia e Innovación of Spain and the University of Vigo for the postdoc grant Juan de la Cierva Incorporación 2019 (IJC2019-042235-I).

**Conflicts of Interest:** The authors declare no conflict of interest. The funders had no role in the design of the study; in the collection, analyses, or interpretation of data; in the writing of the manuscript; or in the decision to publish the results.

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
