# Peer review of "Early Growth Assessment of Lolium perenne L. as a Cover Crop for Management of Copper Accumulation in Galician Vineyard Soils"

_horticulturae, doi:10.3390/horticulturae9091029_

Round 1
Reviewer 1 Report
The researchers studied vineyard areas specifically soil fertility especially copper availability and L. perenne growth. The research discusses the problem of Cu toxicity and how to mitigate this problem using L. perenne. However, in order to have scientific validity, it is important to carry out an in-depth review of the manuscript and, therefore, we have made notes below so that the text has the minimum condition to be accepted for publication. Foliar sprays with copper can increase the copper content in the soil especially near the stem where there is greater flow of the sprayed nutrient solution into the aerial part of the plant. This aspect is important to be discussed in the manuscript. It is relevant to include the soil chemical analysis methodology to determine available copper. Line 397-402 indicates the deficient and toxic levels of Cu, but it is important to detail the extractor used to indicate the available Cu level. In every text when there is a discussion and comparison of the Cu contents of the experiment in relation to the literature, it is relevant to mention the extractor in the text. Plants considered tolerant have a greater ability to accumulate more copper in the roots than in the aerial part of the plant. Therefore, it is important to include the efficiency of Cu transport from roots to shoots. Table 3 indicates the levels of Cu available in the soil and this was discussed to indicate high levels of Cu in some soils collected in the region where the plants are cultivated to reinforce the risks of toxicity in the plants. This discussion is very superficial and scientifically weak if it does not have the results of the foliar chemical analysis. The soil analysis method is an indirect method to assess possible toxicity and the foliar chemical analysis method is a direct method and it should be included in the manuscript to support the discussion of this research.
Author Response
The researchers studied vineyard areas specifically soil fertility especially copper availability and L. perenne growth. The research discusses the problem of Cu toxicity and how to mitigate this problem using L. perenne. However, in order to have scientific validity, it is important to carry out an in-depth review of the manuscript and, therefore, we have made notes below so that the text has the minimum condition to be accepted for publication. Foliar sprays with copper can increase the copper content in the soil especially near the stem where there is greater flow of the sprayed nutrient solution into the aerial part of the plant. This aspect is important to be discussed in the manuscript. It is relevant to include the soil chemical analysis methodology to determine available copper. Line 397-402 indicates the deficient and toxic levels of Cu, but it is important to detail the extractor used to indicate the available Cu level. In every text when there is a discussion and comparison of the Cu contents of the experiment in relation to the literature, it is relevant to mention the extractor in the text. Plants considered tolerant have a greater ability to accumulate more copper in the roots than in the aerial part of the plant. Therefore, it is important to include the efficiency of Cu transport from roots to shoots. Table 3 indicates the levels of Cu available in the soil and this was discussed to indicate high levels of Cu in some soils collected in the region where the plants are cultivated to reinforce the risks of toxicity in the plants. This discussion is very superficial and scientifically weak if it does not have the results of the foliar chemical analysis. The soil analysis method is an indirect method to assess possible toxicity and the foliar chemical analysis method is a direct method and it should be included in the manuscript to support the discussion of this research.
- i) Foliar sprays with copper can increase the copper content in the soil, especially near the stem where there is greater flow of the sprayed nutrient solution into the aerial part of the plant. This aspect is important to be discussed in the manuscript.
Thank you very much for your comment. The study presented uses previously contaminated vineyard soils where there was no vegetation cover. We understand that the application of fungicides can have a direct effect on the growth of Lolium perenne but this was not evaluated in this work, as the soils used are from established vineyards with different Cu contents accumulated over the years.
However, we believe that for future studies we can include an experiment with Lolium perenne and its influence on the foliar application of Cu-based fungicides to study the effect of foliar sprays on this species. In this work, this was not done and we considered studying the effect of Cu already accumulated in the soil, without the influence of the application of the fungicides, in order to be able to know directly the effect of Cu already accumulated in the soil. Therefore, we consider that including it in the discussion section may deviate from the objectives of the paper.
Line 244-256: ii) It is relevant to include the soil chemical analysis methodology to determine available copper
The methodology of the section “2.3. Total and available Cu content in vineyard soils” was modified and corrected.
iii) Line 397-402 indicates the deficient and toxic levels of Cu, but it is important to detail the extractor used to indicate the available Cu level. In every text when there is a discussion and comparison of the Cu contents of the experiment in relation to the literature, it is relevant to mention the extractor in the text.
The available Cu content of in reference 10 was determined by extraction with DTPA (0.005 M DTPA, 0.01 M CaCl2, and 0.1 M TEA adjusted to pH 7.3) in a relation 1 g/10 mL (soil/solution) after shaking for 2 h.
However, we believe that detailing every methodology used in the discussion for all corresponding references is arduous for readers. This is especially true since many values considered phytotoxic are not solely reliant on an extractant, but rather on multiple factors or extractants analyzed. Hence, we assume that readers have access to the relevant reference and can consult the methodology provided in each referenced item.
- iv) Plants considered tolerant have a greater ability to accumulate more copper in the roots than in the aerial part of the plant. Therefore, it is important to include the efficiency of Cu transport from roots to shoots.
Root contents were not determined in this study since the root system of Lolium perenne has a fine structure, and separating the roots from the soil is complicated. Moreover, this would introduce errors associated with the difficulty of separating all the soil particles from the root system. Therefore, we chose to analyze the Cu content using a method that determines the Cu available in the rhizosphere. This enables the values acquired by this methodology to be deemed as absorbable by the roots. It should be noted that the available Cu content extracted with the rhizo method represents the potential Cu that can be absorbed by the roots of this species.
- v) Table 3 indicates the levels of Cu available in the soil and this was discussed to indicate high levels of Cu in some soils collected in the region where the plants are cultivated to reinforce the risks of toxicity in the plants. This discussion is very superficial and scientifically weak if it does not have the results of the foliar chemical analysis.
Thank you for your comment. However, the study was conducted on vineyard soils in Galicia that have been treated with fungicides over the course of several years. Therefore, the data on the content of vineyard leaves is not considered relevant due to the variability in the amount of fungicides used in each vineyard, which is dependent on various factors both within and between years. We do not believe that the Cu content in leaves of vineyards at the time of soil sampling is pertinent to this study, and it falls outside of the project's scope.
However, we did examine the Cu content in the aerial portion of Lolium perenne, which is presented in Table 7 - "Dry biomass production (mg kg-1) of L. perenne, Cu content in aerial part and bioaccumulation factor (BF)".
- vi) The soil analysis method is an indirect method to assess possible toxicity and the foliar chemical analysis method is a direct method and it should be included in the manuscript to support the discussion of this research.
Thank you for your comment. However, the purpose of this study is not to assess the toxicity of Cu on vineyards but rather to identify more effective management strategies using Loliun perenne as a cover crop. The aim is to find new strategies to decrease the fungicide application and Cu accumulation in soils, which may have negative impacts on other species or organisms in the soil, as well as potentially harmful effects on other parts of the ecosystem. Therefore, the leaf analysis of the vines is not included in this research. It is worth noting that half of the soil samples were collected from non-vine areas within the vineyard halls.

Reviewer 2 Report
I have carefully reviewed the manuscript entitled "Lolium perenne L. as a potential cover crop for the improvement of Galician vineyard soils with different Cu contents" submitted by Vázquez-Blanco et al. to the journal Horticulturae. This study aimed to study the effect of different soil physicochemical characteristics on Cu availability and the growth of Lolium perenne, and also to assess the capacity of Lolium perenne to be used as a cover crop and to improve the conditions of Galician vineyard soils under stress caused by high Cu concentrations (using pot experiment). The manuscript is well-written and provide very adequate data. The data involved will help to expand the understanding of phytoremediation in Cu-contaminated soil. This study also provides insight into the potential of Lolium perenne as a cover crop for sustainable vineyard management and soil improvement. For this manuscript I have only one suggestion as follows:
In this study, a total of 42 vineyard soils were sampled from 34 different vineyards. I suggest dividing these soil samples into two groups, namely, low Cu concentration and high Cu concentration (based on the reference limits established for ecosystem protection or phytotoxicity). Then, it is required to statistically compare the differences in physicochemical properties of these two groups of soils with different Cu contents and the responses of Lolium perenne to each group soil.
Author Response
I have carefully reviewed the manuscript entitled "Lolium perenne L. as a potential cover crop for the improvement of Galician vineyard soils with different Cu contents" submitted by Vázquez-Blanco et al. to the journal Horticulturae. This study aimed to study the effect of different soil physicochemical characteristics on Cu availability and the growth of Lolium perenne, and also to assess the capacity of Lolium perenne to be used as a cover crop and to improve the conditions of Galician vineyard soils under stress caused by high Cu concentrations (using pot experiment). The manuscript is well-written and provide very adequate data. The data involved will help to expand the understanding of phytoremediation in Cu-contaminated soil. This study also provides insight into the potential of Lolium perenne as a cover crop for sustainable vineyard management and soil improvement. For this manuscript I have only one suggestion as follows:
Lines 423 and 447: i) In this study, a total of 42 vineyard soils were sampled from 34 different vineyards. I suggest dividing these soil samples into two groups, namely, low Cu concentration and high Cu concentration (based on the reference limits established for ecosystem protection or phytotoxicity). Then, it is required to statistically compare the differences in physicochemical properties of these two groups of soils with different Cu contents and the responses of Lolium perenne to each group soil.
Thank you for your suggestion. We have included the results of the analysis you are asking for in the manuscript but the results were not very different from those obtained with the other correlation analysis and do not provide new information concerning the PCA analysis.

Reviewer 3 Report
Dear Editor
MDPI Agronomy
I am sharing my considerations about the manuscript “horticulturae-2589988 Lolium perenne L. as a potential cover crop for the improvement of Galician vineyard soils with different Cu contents”. The manuscript showed some chemical and physical characteristics (pHH2O, pHKCl, C, OM, N, C/N, DOC, DIC, Ca2+, Mg2+, K+, Na+, Al3+, eCEC, sand, silt, and clay) and Cu contents (total Cu, available Cu and ExEf) in soils from Galician vineyard. A trial in climatic chamber was realised with Lolium perenne by 1 month. This little time is not adequate for the affirmations and conclusions showed in the manuscript. Furthermore, there is no experimental design. The manuscript is a descriptive article about Cu conditions in soils from sampled sites (Galician vineyard). The authors should change the title. Some comments are in the manuscript.
Best regards!
21 August 2023.

Author Response
I am sharing my considerations about the manuscript “horticulturae-2589988 Lolium perenne L. as a potential cover crop for the improvement of Galician vineyard soils with different Cu contents”. The manuscript showed some chemical and physical characteristics (pHH2O, pHKCl, C, OM, N, C/N, DOC, DIC, Ca2+, Mg2+, K+, Na+, Al3+, eCEC, sand, silt, and clay) and Cu contents (total Cu, available Cu and ExEf) in soils from Galician vineyard. A trial in climatic chamber was realised with Lolium perenne by 1 month. This little time is not adequate for the affirmations and conclusions showed in the manuscript. Furthermore, there is no experimental design. The manuscript is a descriptive article about Cu conditions in soils from sampled sites (Galician vineyard).
Line 2: i) The authors should change the title.
We have changed the title according to the manuscript. “Early growth assessment of Lolium perenne L. as a cover crop for management copper accumulation in Galician vineyard soils” We have introduced the early growth concept to adjust the title to study the and the experimental design.
- ii) Some comments are in the manuscript.
Line 192-195: Would be this? or Did you characterize soils in Galician vineyard?
We have rewritten the objectives of the study to adapt them to the results. Regarding soil characterization, all soils sampled were characterized as indicated in the material and methods section.
Line 259-261: Why did you adopt only 1 month to plants in climatic chamber?
Thank you for your question.
The main objective of the study was to evaluate the tolerance and accumulation of Cu by Lolium perenne in vineyard soils. We have changed the title and objectives to be more specific about "early growth". A growth period of one month was chosen as the initial duration to make a preliminary assessment of the plant response to Cu, mainly on the initial growth of this species, and to provide a solid basis for future research in field trials.
One month of growth may be sufficient to get an idea of the response of Lolium perenne to soil conditions with different Cu contents in a controlled experiment in a climate chamber. Short-term responses can provide information on how the plant interacts with copper and how it responds at the physiological and accumulation levels. One month of growth allows initial trends in Cu accumulation, growth and overall response to be observed in Lolium perenne, which grows rapidly during this period and allows sufficient biomass to be obtained to assess aerial Cu levels. It may also be important to assess the early response of Lolium perenne to understand how this species initially adapts or responds to copper stress in a controlled environment.
Line 282: What was experimental design? Would be entirely at random?
Thanks for the question. It was incorrectly worded in the text. It has been corrected.
Line 447: Could be this result different if the experimental time in climatic chamber would be longer than one month?
Yes, a longer experimental period could reveal possible trends or effects that may not be apparent in the initial one-month time frame. Some biological processes, such as gradual adaptation or changes in nutrient availability due to soil-copper interactions, may require more time to manifest and could influence the response in L. perenne. With prolonged exposure, interactions between L. perenne roots and soil and processes in the rhizosphere could influence Cu uptake. Longer exposure could lead to redistribution of Cu within the plant as it grows, possibly affecting different parts of the plant differently. This could affect the total biomass and provide a more complete understanding of the plant response.
In any case, the main objective of the study was to gain an initial understanding of the response of L. perenne to soil Cu levels over a relatively short period. This choice was based on the need to quickly assess whether there was evidence of significant plant-copper interactions, particularly in the early stages of growth. The one month period was chosen to observe the direct and more immediate response of plants to different levels of copper. This could help to identify initial trends and differences in growth in a narrow time frame. The results of the study can be seen as a starting point for further research. While the one-month period may not reveal all long-term effects, it can provide a platform for designing more extensive and detailed field studies, as we are currently doing.

Round 2
Reviewer 1 Report
The manuscript was improved and the authors' responses explained critical points of the work and can be accepted for publication.